# Predicting climate-change induced heat-related illness risk in Grand Canyon National Park visitors

Danielle E. Buttke[1]*, Brinkley Raynor[2], Gregor W. Schuurman[1]

1 Natural Resource Stewardship and Science Directorate, National Park Service, Fort Collins, Colorado, United States of America, 2 Department of Biostatistics, Epidemiology, and Informatics, University of Pennsylvania, Philadelphia, Pennsylvania, United States of America

* danielle_buttke@nps.gov

## Abstract

### Background

The climate crisis is the greatest public health threat of the 21st century. Excessive heat is responsible for more deaths than any other extreme weather event, and the frequency, intensity, and duration of extreme heat events are increasing globally due to climate change. Exposure to excessive heat can result in heat related illnesses (HRIs) and long-term poor health outcomes. Physical exertion, sudden exposure to excessive heat, and the lack of physical or behavioral adaptation resources are all associated with greater HRI risk, which is expected to increase for visitors to Grand Canyon National Park (GCNP) and other public lands as climate change worsens.

### Objectives

Our objectives were to understand 1) the relationship between weather and HRI in GCNP visitors, 2) how future HRI rates may change, and 3) how land management agencies can update risk mitigation strategies to match changing risk and better manage an increased HRI burden.

### Methods

We utilized previously published data on HRI in GCNP visitors, and records of daily visitation, temperatures, and maximum and minimum daily humidity from the same study period to develop a model estimate for HRI risk. We then used future climate projections from the World Climate Research Programme's Coupled Model Intercomparison Project phase 5 multi-model dataset to model future HRI risk under different climate scenarios.

### Results

The incidence of HRI was significantly associated with maximum daily temperature and minimum relative humidity, and was more common in the shoulder season months. We

**Data Availability Statement:** All relevant data are within the paper and its Supporting information files.

**Funding:** The authors received no specific funding for this work.

**Competing interests:** The authors have declared that no competing interests exist.

estimated that HRI will increase 29%-137% over 2004–2009 levels through 2100, assuming no change in visitation.

## Discussion

Climate change will continue to increase HRI risk for GCNP visitors and poses risks to public land managers' mission to provide for safe recreation experiences for the benefit of this and future generations in places like GCNP. Excessive risk during the shoulder season months presents an opportunity to increase preventative search and rescue and education efforts to mitigate increased risk.

## Introduction

Climate change is the greatest public health threat of the 21st century [1]. Negative public health implications of climate change include reduced air quality and access to clean water, increasing malnutrition, vector-borne, and diarrheal diseases, increasing civil conflict and migration, and negatively impacted mental health [2, 3]. Excessive heat is responsible for more deaths than any other extreme weather event, and the frequency, intensity, and duration of extreme heat events are increasing globally due to climate change [2, 4, 5]. Heat-related illness (HRI) includes acute conditions of heat cramps, heat exhaustion, and heat stroke, but can also exacerbate underlying health conditions such as respiratory or heart disease, diabetes, or cerebrovascular disease [6–8]. High ambient temperatures have significant negative impacts on mental health, including a significant increase in suicide risk and violent crime, and can even hinder learning ability, productivity, and cognitive performance [9–12]. Those who recover from HRI face increased risk of cardiovascular and chronic disease and early death [13–16]. Like many other aspects of climate change, the burden of excessive heat falls disproportionately on disadvantaged and minority communities, who also have fewer adaptation and recovery resources [5, 17, 18].

National parks and other public lands are important public health resources where people increasingly seek the physical-, mental-, and social-health benefits of nature [19]. However, the nature of people's use of public lands also poses distinct climate change-related health risks because 1) visitors often engage in more strenuous exertion and experience greater exposure to the elements with fewer resources than experienced in daily life, and 2) these areas are increasingly experiencing unprecedented climate and weather conditions that may surprise experienced visitors and even park staff [17, 20, 21]. Warming temperatures, including increasing frequency of very hot days, may be the most consistent expression of ongoing climate change across the U.S. National Park System—a vast array of over 400 management units centered in North America but spanning the globe from the south and west Pacific to the Arctic and the Caribbean. Most U.S. national parks are at or beyond the extreme warm edge of their historical ranges of variability [21]. Unanticipated heat during historically cooler months and in park units with historically milder climates increases the risk of HRI for unprepared visitors. Parks such as Grand Canyon National Park (GCNP) with a history of HRI are therefore increasingly modifying their approaches in light of recent increases in rates, and parks such as Yosemite National Park in California and New River Gorge National Park and Preserve in West Virginia that have not historically encountered the issue are developing/adopting mitigation approaches (e.g., [22]).

Providing safe and healthy visitor experiences is an important part of the U.S. National Park Service (NPS) mission. Understanding conditions associated with increased HRI is

critical for anticipating and managing this risk, both in the near term and in longer climate change adaptation planning and management of visitor services. Common NPS visitor HRI symptoms include heat exhaustion, heat stroke, and hyponatremia (low blood sodium), and these events result in visitor hospitalizations and deaths every year [20, 23]. Although the creation of a preventative search and rescue (SAR) program in GCNP in 1997 substantially improved health outcomes and reduced emergency medical services (EMS) HRI responses, staffing for this program is limited to the peak visitation season, which may not mirror peak HRI risk [24, 25]. Additionally, most public lands lack these resources and do not have a data-informed model to appropriately target limited resources to best affect public health outcomes. As temperatures rise and patterns of extreme heat become more variable, preventative SAR resources may not be ideally structured to address changing risk patterns nor increasing demand, and GCNP park EMS and preventative SAR services are again at risk of becoming overwhelmed by demand. At the same time, public lands lacking these HRI mitigation resources would benefit from greater understanding of how these programs might be developed to both understand and address peak risk intervals.

By integrating historical NPS HRI data and current NPS mitigation protocols and climate projections, we aim to better understand 1) the relationship between weather, human behavior, and HRI, 2) how future HRI rates may change, and 3) how land management agencies can update risk mitigation strategies to match changing risk and better manage an increased HRI burden. This information will allow the development of adaptive SAR and education programming targeting the highest risk conditions to improve public health outcomes as well as improve public land managers' understanding of how climate change threatens their mission to provide for safe recreation experiences for the benefit of this and future generations.

## Materials and methods

### Study site

Our study site, GCNP provides a unique opportunity to understand HRI in a national-park context because its well documented history of individual HRI incidents and reliable information about visitor numbers provide a unique opportunity to understand HRI in a national-park context. In 2018, visitor numbers to GCNP reached a peak of 6.38 million people with most visits occurring during the peak season months [26]. Every year, park visitors unprepared for the intense exertion and high temperatures within the canyon suffer from HRI events; particularly because conditions at the top of the canyon where visitors begin their hikes are often much cooler than conditions at the bottom of the canyon, where visitors also transition from hiking downslope to having to hike up a considerable incline (Fig 1; [20, 23, 24]).

### Data

All data used on our study are publicly available either online or provided by the study authors of a previous publication with permission granted for use and publication. A previously anonymized and published dataset of GCNP HRI case reports from 2004–2009 [23] was provided by the study authors and used to build the outcome of interest (Table 1). Both paper and electronic medical records were reviewed by a trained public health officer and records meeting the case definition were extracted for the study. HRI events were defined as:

"Persons whom EMS treated (or found if deceased) were eligible for inclusion into the study if they were a visitor during the study period and had a documented outdoor heat exposure of more than 1 hour within the previous 24 hours. To be a case there need to be one of the following documented indicator(s) of HRI: 1) paramedic assessment of HRI; 2)

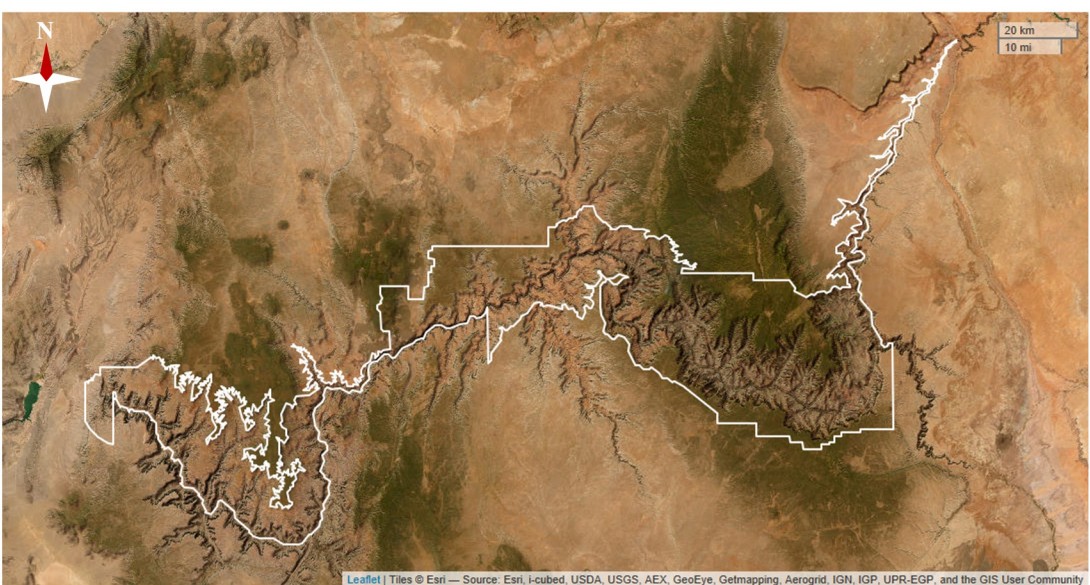

**Fig 1. Map of the study site with Grand Canyon National Park boundaries depicted in white.** Park boundary data are publicly available through NPS IRMA [26] and base map data is publicly available [27].

self-report or phrases that indicated "heat" (e.g., "overheated, dehydration") as a contributing factor; or 3) at least 2 signs and symptoms consistent with HRI not associated with another etiology (e.g., nausea and vomiting while hiking)" [23].

Only records with complete data on the date of the event, paramedic assessment or self-reported symptoms, location, and reported activity were included in the study.

We estimated daily visitor numbers by first summing estimates of automobile-based visitor numbers from the park's three automobile entry points—the South Rim, North Rim and Desert View visitor stations [28]—and then adding to this sum an estimate of daily visitors arriving by other means. The vast majority of park visitors (>75%, NPS [unpublished]) arrive in automobiles, for which the NPS provides daily visitor estimates. However, small proportions arrive by other means, including by bus (<20%), train (<5%), and boat (<<1%), for which

**Table 1. Data type, purpose, and source used in a predictive model of heat illness risk in visitors to Grand Canyon National Park, 2004–2009.**

| Data type | Purpose | Source |
|---|---|---|
| HRI events | Regression outcome | Noe et al, 2013 |
| Visitation records | Regression predictor | GCNP visitor use statistics program |
| Historical climate data (recorded daily $T_{min}$, $T_{max}$, $RH_{min}$, $RH_{max}$) | Regression predictor | GridMET: gridded daily surface meteorological dataset (Abatzoglou, 2013) |
| Projected climate data (predicted daily $T_{max}$, $RH_{min}$) from 14 GCMs (global climate models) | Prediction input | World Climate Research Programme's Coupled Model Intercomparison Project phase 5 (CMIP5) multi-model dataset (Taylor et al., 2012), statistically downscaled using the MACA (Multivariate Adaptive Constructed Analogs) method (Abatzoglou & Brown, 2012)." |

only monthly counts exist. For this study, we distributed these monthly counts evenly across the days in the month, adding them to the automobile-based visitor-count estimate for each day to obtain a daily total park visitor estimate. NPS social science program staff suggested this approach because most such visitors are international travelers, a group whose visitation is not strongly influenced by day of the week or month.

Historical climate data, including daily minimum temperature (Tmin), maximum temperature (Tmax), minimum relative humidity (RHmin), and maximum relative humidity (RH max), were extracted from the Abatzoglou gridded daily surface meteorological (gridMET) dataset, for the period from 1979 to 2018 [29]. We used data from the 4-km gridMET grid cell that includes Grand Canyon Village (Lat 36.054390/Lon -112.140143) in the park's heavily visited South Rim District. We calculated heat index using the standard methods described by the U.S. National Oceanographic and Atmospheric Administration [30]. We chose daily $T_{max}$, daily $RH_{min}$, and time of year (month) as our predictor variables based on a strong correlation between these indices and HRI and death [6].

To characterize how HRI incidence might change in the park through the rest of the 21st century, we obtained daily $T_{max}$ and $RH_{min}$ data (projections) for the period 2018–2098 from 14 global climate models (GCMs) in the World Climate Research Programme's CMIP5 (Coupled Model Intercomparison Project phase 5) multi-model dataset [31], statistically downscaled using the MACA (Multivariate Adaptive Constructed Analogs) method [32]. The MACA method uses training data from gridMET and therefore MACA-based projections are compatible with gridMET records. We obtained these data for the visitation season of interest (April 1-Sept 30) from the 4-km MACA grid cell that includes Grand Canyon Village (Lat 36.054390/Lon -112.140143) in the South Rim District. We worked with climate uncertainty by considering a range of plausible climate futures [33, 34] represented by 28 projections. These 28 projections consisted of two simulations each of 14 downscaled CMIP5 global climate models, in which one simulation used a moderate greenhouse gas emissions pathway that assumes lower future emissions rates due to technological advancements and policy change (Representative Concentration Pathway [RCP] 4.5) and the other used a high emissions pathway (RCP 8.5) [35]. Pathways of GHG emissions have been standardized as a set of Representative Concentration Pathways (RCPs; [35]) that reflect the climate consequences of a broad range of socio-economic futures and have been accepted by the international community. A large set of GCMs, reflecting different quantitative representations of atmospheric dynamics and different sets of Earth system processes, have used these standardized RCPs to generate climate projections through the twenty-first century.

## Data cleaning

We compared the original dataset to preventative SAR reports from the GCNP Visitor and Resource Protection Office and compared deaths in our data set with public records to ensure completeness. We imputed missing visitor numbers from any individual visitor station on a given day using the MICE package in R and the CART method due to the high degree of correlation between visitor numbers from all other stations and date, with daily visitor numbers from all other visitor stations and day of the year as predictors [36].

Our primary outcome was the number of HRIs per week per estimated 100,000 visitors during the peak season. The peak season here is defined as April 1 through September 30. Therefore, we binned the data (structured by date) into weeks, with weeks starting on Sunday and ending on Saturday. Recorded HRIs and recorded visitation were summed per week, and temperature and humidity data were averaged across the week. Month was assigned per week based on which month the majority of the week fell into.

## Regression model development

We built general linear regressions predicting the number of heat events per estimated 100,000 visitors per week based on climate variables including $T_{max}$, $RH_{min}$, heat index, year, time of year, and number of visitors. Following methods described in Hosmer and Lemeshow [37], we used a forward selection method to build a suite of reasonable negative binomial and Poisson models (S1 File). Our final model selection was based on a minimum Akaike Information Criterion [38] and significantly improved performance via a likelihood ratio test [39].

We applied the regression predictions to climate projections from 14 different global climate models and then fitted a linear model to the projections relating HRI events to year. (The term 'projection' refers to a climate model [e.g., HadGEM2-CC365] driven by a specific radiative forcing [e.g., RCP 4.5] Fig 3, S2 File).

## Statistical analysis

We used R for all statistical analysis [40]. We used the "stats" package generalized linear model function to construct Poisson models and the negative binomial generalized linear model function in the "MASS" package to construct negative binomial regressions [41]. We made the map using the "leaflet" package and all figures using the "ggplot2" package [42].

## Results

For the years 2004–2009, estimated average weekly peak season visitation to Grand Canyon National Park during the months of April 1 to September 30 was 110,596 individuals, with a range of 13,486 to 153,560. Peak visitation occurred in the month of July. A total of 483 HRI events was recorded, including 6 deaths. The average weekly number of HRI events per 100,000 visitors per week during our study period was 2.68 events, with the highest number of HRIs (12 cases) occurring in July 2005 and no HRIs recorded in 23 out of 120 (19%) weeks over the study period.

The best-fitting model was a Poisson regression with the variables of $RH_{min}$, $T_{max}$, month (nominal), and month-$RH_{min}$ interaction (Table 2). We plotted predicted heat events against

**Table 2. Variables, model estimates, standard error, and p-values of parameters evaluated for inclusion in the final regression model predicting the rate of heat related illness events in Grand Canyon National Park visitors, May through October, 2004–2009.**

| Variable | Estimate | Standard error | *p*-value |
|---|---|---|---|
| Intercept | -11.85 | 1.21 | <0.01 |
| Average weekly maximum temperature ($T_{max}$) | 0.03 | 0.01 | 0.04 |
| Average weekly minimum relative humidity ($RH_{min}$) | -0.1 | 0.05 | 0.04 |
| May (compared to April reference) | 0.09 | 0.75 | 0.91 |
| June (compared to April reference) | -0.28 | 0.8 | 0.72 |
| July (compared to April reference) | -0.27 | 0.77 | 0.73 |
| August (compared to April reference) | -0.88 | 0.81 | 0.28 |
| September (compared to April reference) | -2.3 | 0.92 | 0.01 |
| Interaction of $RH_{min}$ and May | 0.06 | 0.05 | 0.28 |
| Interaction of $RH_{min}$ and June | 0.04 | 0.06 | 0.54 |
| Interaction of $RH_{min}$ and July | 0.04 | 0.05 | 0.4 |
| Interaction of $RH_{min}$ and August | 0.08 | 0.05 | 0.15 |
| Interaction of $RH_{min}$ and September | 0.17 | 0.06 | <0.01 |

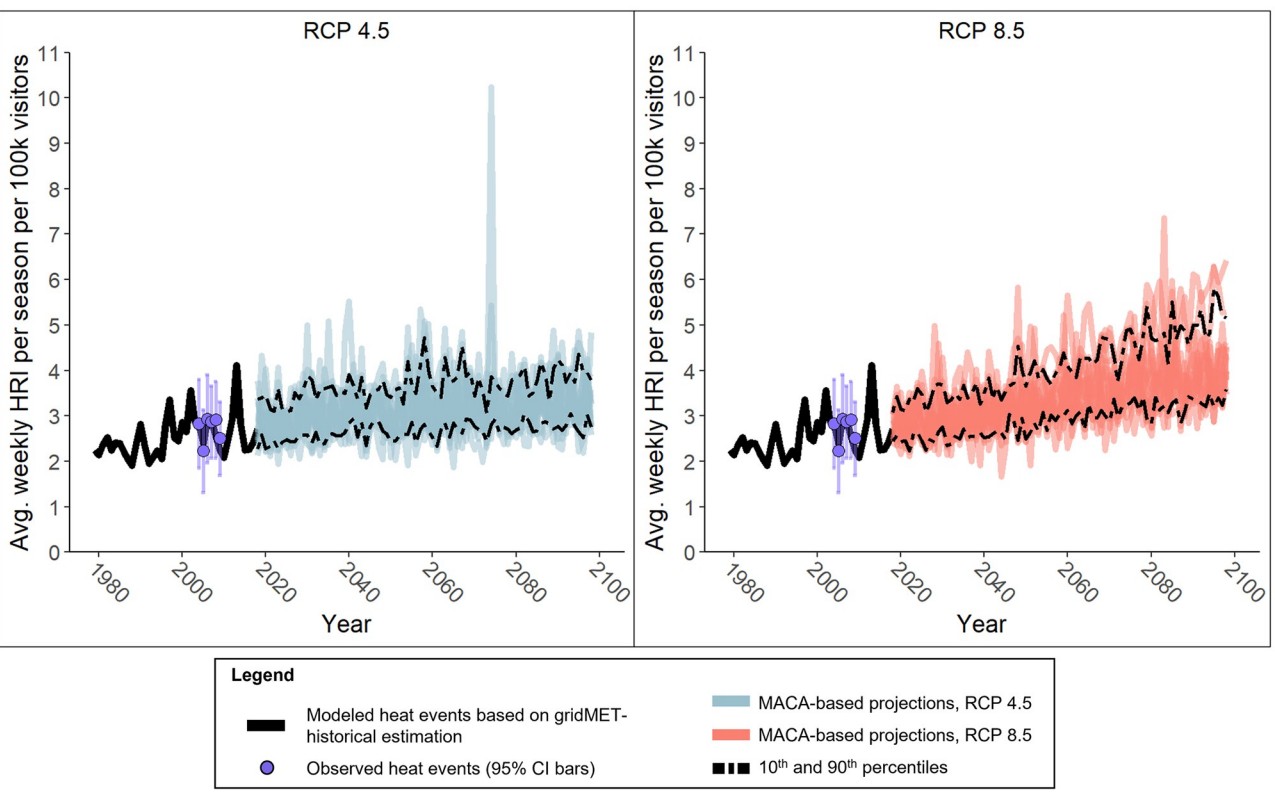

**Fig 2. Modeled heat events per 100,000 visitors in Grand Canyon National Park from 1979 through 2100.** A Poisson regression was constructed using observed heat events in GRCA from 2004–2009. Represented here is the daily average per each year of both the modeled (black line) and each year's daily average observed events (purple points) and its 95% confidence interval (purple error bars). Additionally, the regression model was applied to 14 global climate models and projected heat events are shown in blue (RCP 4.5) and red (RCP 8.5) for each model with the 10th and 90th percentile of the ensemble marked with a dotted line.

recorded heat events in Fig 2 to test model performance. Fig 3 shows the projected increase in mean daily heat events for RCP 4.5 and RCP 8.5 for 14 global climate models.

Our model predicted an increase in the weekly average peak season HRI events per 100,000 visitors for all projection models examined, with a range of 0.0035 (RCP 4.5 CanESM2) to 0.032 (RCP 8.5 MROC-ESM-CHEM). This range equates to a total of 0.10 to 0.96 more weekly HRI events each peak season, assuming visitation remains at 2004–2009 levels. This corresponds to an average increase of 29%-137% over 2004–2009 levels, assuming no change in visitation. However, average annual visitation to GCNP is increasing—the estimate for the six-year period on which this study focused (2004–2009) was 4,365,707.5 visitors/year whereas the estimate across six years a decade later (2014–2019) was 5,809,410, an increase of over 33%. If we assume 2014–2019 visitation levels in 2100, 138–254 total peak season HRI events would be expected for RCP 8.5 models, and that assumption may also ultimately prove to be an underestimate because visitor numbers may increase further [43].

## Discussion

### Heat projection

We found that increased temperatures and decreased relative humidity increased HRI risk. The expected number of HRIs increases by a multiplicative factor of 0.03 for every 1-degree-

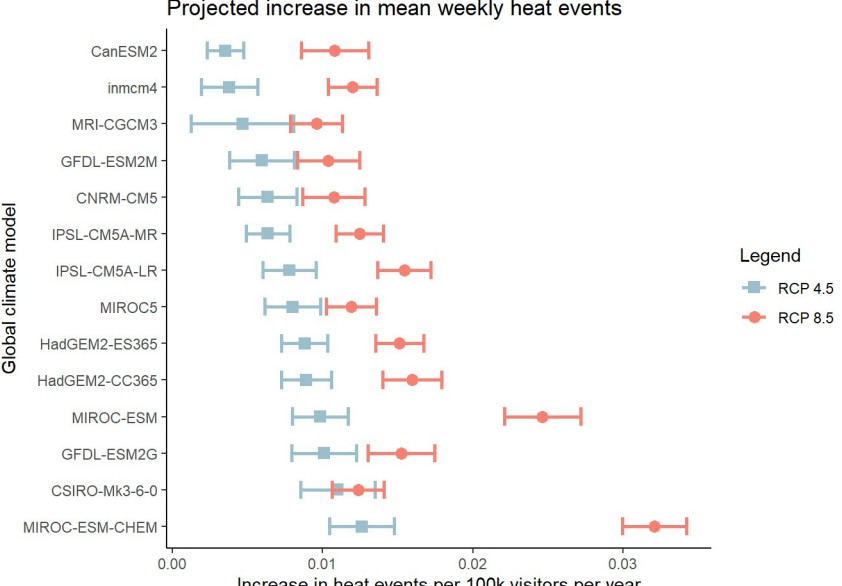

**Fig 3. Projected increase in mean weekly heat related illness events per 100,000 visitors to Grand Canyon National Park for representative concentration pathway (RCP) 4.5 (blue) and RCP 8.5 (red) for 14 global climate models from the World Climate Research Programme's Coupled Model Intercomparison Project phase 5 multi-model dataset.**

Fahrenheit increase in maximum temperature with all other factors being equal, and decreases by a multiplicative factor of 0.9 for every 1-unit increase in minimum relative humidity with all other factors being equal. We also found increased risk associated with perceived cooler peak-season months (April and May) when visitors may not be as prepared for HRI. This finding is consistent with findings of [23, 44], who attributed this effect to lack of behavioral adaptation to unexpected heat, which is more likely to occur during the shoulder seasons and is a critical focus for reducing morbidity and mortality from HRI [45]. Finally, we found a significant interaction between month and $RH_{min}$, suggesting that the effect of very dry conditions (minimum relative humidity) varies across months, again perhaps due to visitors' expectations, water packing, and exertion. Prior studies have also found a close association between physical exertion and HRI as well as a change in the climate zone and wearing clothing that is not suitable for the climate, which is common with air travel to national parks, as significant risk factors for HRI [46]. Behavioral adaptation is key to reducing mortality from extreme heat events but requires planning and resources that national park visitors may not be prepared to use without additional awareness and education efforts [47, 48].

HRIs increased significantly under all climate projections. Recent work has found RCP 8.5 models most closely model historical cumulative emissions as well as match most closely with projected emissions based on current policy through the mid-21st century [49]. These results suggest the RCP 8.5 models may more closely predict the actual increase in HRI most likely to be seen at GCNP and predict a concerning increase in disease burden among visitors.

## Management implications

The Center for Disease Control and Prevention's Climate and Health Technical Report identifies health communication as a primary intervention to decrease HRI risk [17]. At GCNP, alerts and warnings are published at NPS.gov/GRCA, on signs throughout the park, and on

social media and virtual communication channels. These communications will likely remain a major part of HRI prevention activities for the NPS. Park managers have also made structural changes in recent years, such as building more shade structures. As temperatures increase in coming decades, more such structures may be needed, especially along popular inner-canyon trails (North and South Kebob trails and Bright Angel trails). Another important resource along these trails is potable water. There are several water stations along these inner-canyon trails and a team of rangers called the preventative SAR team provides water and assistance along popular trails, but water availability is inherently limited in the natural, back-country, desert environment found in the Grand Canyon [24]. Similarly, air-conditioned cooling centers are not available in these remote locations nor is there sufficient space in air-conditioned spaces in developed areas of the park to serve all visitors during the peak season months [17]. Occupational heat exposure poses an even greater risk for outdoor workers with physical exertion during heat exposure, which includes many NPS employees such as preventative SAR rangers [50]. As temperatures continue to rise and resources in these back-country locations remain limited, certain previously-used parts of national parks and public lands may not be safe for recreation or stewardship operations during periods of extreme heat. As the climate continues to warm, these periods of extreme heat will become longer, more intense, and more frequent. The NPS is strongly committed to considering climate change adaptation in all its planning processes, but significant changes in important places and loss of access to many important resources may occur as the climate crisis worsens and recreation becomes less safe in these remote settings.

HRI is perhaps the most obvious direct impact of warming temperatures [5, 17]. Some parts of the globe are expected to cease be habitable to humans due to elevated ambient temperatures, so heat is also viewed as a direct driver of climate-induced migration and civil conflict [5, 17, 51]. However, other significant indirect health impacts may also be detrimental to public health [5, 17]. Increased frequency of extreme weather events and wildfires take a dramatic toll on human life and mental health, and declines in agricultural productivity due to climate change are projected to threaten global food security in all regions. Rising temperatures and increased variability in precipitation can dramatically increase the potential for vector-borne and waterborne disease transmission. National parks are highly visited and loved resources where visitors are more likely to be open to new information, behaviors, and transformative experiences [52, 53]. Additionally, exposure to extreme heat or prior HRI events is associated with higher levels of behavioral adaptation and concern about heat risks [54]. Becoming aware of and managing the risk of extreme heat impacts during a recreational visit to a park may therefore have significant potential to stimulate climate change adaptation and mitigation behaviors in visitors elsewhere after their visit, and therefore national parks may have an opportunity for sustained impact on public ethics and behaviors by highlighting climate change in HRI prevention messaging.

Previous studies have found that the greatest burden, most severe disease, and highest overall expenses for treatment of HRI fall disproportionately on minorities and older populations, worsening existing inequities and disproportionately burdening publicly funded systems [55]. Populations without access to air conditioning or with inadequate access to water or health services are more vulnerable to adverse health outcomes from exposure to excessive heat, conditions that can occur from a lack of development or when temperatures rise faster than communities are able to adapt, resulting in a significant disproportionate burden on disadvantaged and minority communities impacted by structural racism [5, 17]. Visitors to the U.S. national parks are disproportionately white, educated, and affluent compared to the general U.S. population, and therefore less likely to be as directly impacted by the earliest impacts of climate change. By highlighting the impacts of climate change to national park visitors, the NPS

therefore may engage broader audiences in the conversation about climate change impacts on human health and the need for adaptation and mitigation than through approaches that focus on personal impacts alone.

## Limitations

This study is subject to several limitations. It uses data from 2004–2009, after a preventative SAR program was developed by the park and after significant warming had already occurred, which may bias our projections downwards. Other areas where such harm-reduction measures have not been put in place are likely to have a more significant burden of disease than that predicted by our model. The study also likely underestimates HRI events because not all affected individuals sought health care and some victims may have sought care at health care facilities outside of the park, such as at local health care facilities in gateway communities, and/or waited to seek care until they returned home. We also cannot account for adaptation, neither behavioral nor physiological. Although physiological adaptation may be less feasible in non-local visitors, behavioral adaptation is likely to occur, particularly as extreme heat becomes more common. GCNP established a preventative SAR program in 1996, which provides support staff with food and water to prevent adverse health outcomes in visitors, and this program significantly reduced heat-illness related SAR responses and highlights the success of behavioral adaptation in preventing HRI [24].

The range of climate projections used in this study yielded a 10-fold difference in predicted HRI burden. We cannot know today which projection will ultimately turn out to be the most accurate because we cannot know either precisely 1) what future emissions choices humans will make or 2) which GCM among the set we used for our climate projections will most skillfully characterize how climate in this region responds to those future emissions. Therefore, we present the full set of projections, so that managers can understand and address the full range of plausible outcomes [34]. However, as noted above, RCP 8.5-based models characterize historical emissions most closely and are consistent with current emissions-related policy, and thus likely provide the best approximation of future conditions [49] unless humans reduce greenhouse gas emission rates substantially and soon. Our model also cannot address the interaction of heat and air quality. Heat impacts air quality by increasing production of both ground-level ozone and particulate matter (through increased evapotranspiration), and health impacts from poor air quality can mimic and worsen some signs and symptoms of HRI [56]. Additionally, this model is based only on heat events occurring during the current main season at GCNP from April through September and therefore cannot predict any change in events occurring outside of this season. Our model also does not evaluate the impact of rising minimum temperatures, which can play an important role in HRI as physiologic night-time cooling is impaired [57]. We are currently working to expand this work into additional months and contemporary years to better understand HRI risk. Finally, our study focuses exclusively on GCNP visitors and may not be useful for predicting future HRI in other populations whose behaviors may differ significantly from those of the recreating public or for public lands with fewer resources for prevention and response. Specifically, other public lands may see more significant increases in HRI due to the presence of fewer resources for prevention and response, particularly the lack of a preventative SAR program focused on HRI prevention.

## Conclusions

Rates of HRI will increase significantly in GCNP visitors in coming decades, with shoulder-season extreme heat increasing heat illness risk more than mid-season extreme heat, as well as extreme heat occurring during low humidity. Because RCP 8.5 climate models most accurately

track historical and current warming trends, the highest heat illness risk projections are the most likely and make heat illness risk a significant need for public land management planning and staffing efforts. Like many other aspects of climate change, extreme heat is a threat to human health and wellbeing and poses a significant challenge for public land managers to carry out the mission to provide for safe visitor experiences in our national parks. This threat will worsen with time regardless of how quickly society acts to mitigate continued warming. Land managers and the visiting public alike should be aware of the significant risk climate change poses to human health and wellbeing and work to mitigate these risks in their daily work and lives.

## Supporting information

**S1 Fig. Predicted heat events vs recorded heat events.** Figure of recorded heat events (black) and predicted heat events (red line) at Grand Canyon National Park from 2004–2009; the pink shaded areas representing 95% prediction confidence intervals from the final negative binomial regression model.
(TIF)

**S1 File. Binned HRI data.** Deidentified binned data of visitation estimates and recorded HRIs used in regressions.
(CSV)

**S2 File. Poisson model AICs table.** Table of AIC of different Poisson models tested.
(CSV)

**S3 File. Predicted HRI increase table.** Table of predicted mean increase in cases per year for each climate model, estimated using lm(Events ~ year). Events increased per year taken as coefficient of slope [95% CI].
(CSV)

**S4 File. R code of regression model fitting.** Rmd file with code and output for regression model fitting.
(HTML)

**S5 File. R code of regression model prediction.** Rmd file with code and output for regression model prediction.
(HTML)

## Acknowledgments

We would like to thank Pam Ziesler (NPS) for assistance with GCNP visitation data; Amber Runyon (NPS) for assistance with climate model data and manuscript review; Joel Reynolds (NPS) for careful review and comments on statistical methods and interpretation; Nick Fudala (NPS) for assistance with interpreting daily visitation data for GCNP; David Lawrence (NPS) for review of data visualization and manuscript; Jennifer Proctor (NPS) for access to the historical Noe et al. 2013 dataset. We also extend our gratitude to the GCNP Preventative and regular Search and Rescue Rangers who protect visitor health and safety.

## Author Contributions

**Conceptualization:** Danielle E. Buttke, Gregor W. Schuurman.

**Data curation:** Danielle E. Buttke, Gregor W. Schuurman.

**Formal analysis:** Danielle E. Buttke, Brinkley Raynor.

**Methodology:** Danielle E. Buttke.

**Project administration:** Gregor W. Schuurman.

**Validation:** Brinkley Raynor.

**Visualization:** Brinkley Raynor.

**Writing – original draft:** Danielle E. Buttke, Gregor W. Schuurman.

**Writing – review & editing:** Danielle E. Buttke, Brinkley Raynor, Gregor W. Schuurman.

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
