## [Decision Letter · Decision Letter 0]

1 Feb 2023

PONE-D-22-35008Climate change will increase heat-related illness risk in Grand Canyon National Park visitorsPLOS ONE

Dear Dr. Buttke,

Thank you for submitting your manuscript to PLOS ONE. After careful consideration, we feel that it has merit but does not fully meet PLOS ONE’s publication criteria as it currently stands. Therefore, we invite you to submit a revised version of the manuscript that addresses the points raised during the review process.

We look forward to receiving your revised manuscript.

Kind regards,

Mohammed Magdy Hamed

Academic Editor

PLOS ONE

Journal Requirements:

2. Please provide a complete list of data sources (publications, databases etc) that were sued in this study.

3. Please confirm that all data sources you used were publicly available and anonymized. If this is not the case, please provide information on what permissions you were granted to access these data.

5. We note that you have referenced (>75%, unpublished NPS data) on page 8 which has currently not yet been accepted for publication. Please remove this from your References and amend this to state in the body of your manuscript: (ie “Bewick et al. [Unpublished]”) as detailed online in our guide for authors

6. Your abstract cannot contain citations. Please only include citations in the body text of the manuscript, and ensure that they remain in ascending numerical order on first mention.

7. Please upload a copy of Supporting Information Figure/Table/etc. Supplemental Table 2 which you refer to in your text on page 11.

Additional Editor Comments:

I have completed my evaluation of your manuscript. The reviewers recommend reconsideration of your manuscript following MAJOR revision and modification.

Reviewers' comments:

Reviewer's Responses to Questions

**Comments to the Author**

1. Is the manuscript technically sound, and do the data support the conclusions?

Reviewer #1: Partly

Reviewer #2: Yes

Reviewer #3: Yes

2. Has the statistical analysis been performed appropriately and rigorously? 

Reviewer #1: Yes

Reviewer #2: I Don't Know

Reviewer #3: Yes

3. Have the authors made all data underlying the findings in their manuscript fully available?

Reviewer #1: Yes

Reviewer #2: No

Reviewer #3: Yes

4. Is the manuscript presented in an intelligible fashion and written in standard English?

Reviewer #1: No

Reviewer #2: Yes

Reviewer #3: Yes

5. Review Comments to the Author

Reviewer #1: Reviews

The study titled “Climate change will increase heat-related illness risk in Grand Canyon National Park visitors" seems an interesting study contributed by authors. However, there is more space for authors to make it more effective for the research community. Therefore, major suggestions regarding this study area as follows:

1.The title of the research paper may be changed.

2.Authors need to write a clear research gap in the introduction section of this research study.

3.There are many grammatical mistakes in the paper. Those cannot be pointed out one by one. So, it is recommended that this paper should be reviewed from English expert for proof reading to make it more attractive.

4.Why gridMET gridded data set was used in this study? You can also try to do this with other gridded datasets such as CRU etc. Furthermore, compare their results with gridMET gridded dataset.

5.Boundaries of study area and total area of the study area may be identified using Google earth engine or ArcGIS in the revised section of the paper.

6.The authors can prepare a table including all type of collected data and GCMs used in the study.

7.The data collection period of 2004-2009 for prediction of HRI is less period. However, it is suggested for the authors to collect more data from 2010 and onwards to make regression model more effective if the data is easily available for authors from literature review or from concerned NPS authorities.

8.Why these 14 GCMs were used in your study? You can review some literature which indicates the best skillful GCMs for the study area and can be used in this study. By doing this, results would be more reliable and authentic.

9.The results of Figure 1 are not clear. Furthermore, Figure 1 results can also be done at 99% CI band to explore more about observed heat events.

11.The conclusion section of this study does not reflect the outcomes of the whole study. Therefore, it should be revised to include major contributions.

Reviewer #2: The article is well-written. The arguments are supported with references. However, some modifications are recommended to enhance the readability of the paper. Overall, consider revising the language in the manuscript for minor errors and the reference style, and add more figures. The resolution of the figures needs to be enhanced with at

least 300 dpi.

Please see the attachment for detailed comments.

Reviewer #3: The paper titled "Climate change will increase heat-related illness risk in Grand Canyon National Park visitors" is a very important and significant work that provides valuable insights into the potential impact of climate change on the health and well-being of Grand Canyon National park visitors. The study presents a clear and well-structured argument, supported by a thorough analysis of the data, and provides a strong foundation for future research in this area. One of the strengths of the paper is its descriptive nature, providing a detailed account of the study methods and results. The use of statistical models and the careful consideration of the limitations of the data and methods is commendable. However, it would be beneficial for the manuscript to include more formal explanations of certain concepts, such as the standard heat index equation used in the study. Additionally, it will be more informative if the authors provide more details about the study area and its characteristics.

6. PLOS authors have the option to publish the peer review history of their article (what does this mean?). If published, this will include your full peer review and any attached files.

Reviewer #1: No

Reviewer #2: No

Reviewer #3: No

---

## [Author Response · Author response to Decision Letter 0]

5 Apr 2023

We have formatted our files in accordance with the PLOS One journal formatting guidelines. The data used in this study were publically available, and all data have now been uploaded in a compressed zip file labeled as Supporting Information. We have amended our data availability statement to address the uploaded Supporting Information, and with the upload of the Supporting Information, the referenced 'Supplemental Table 2' is now uploaded and available.

---

## [Decision Letter · Decision Letter 1]

5 Jul 2023

Predicting climate-change induced heat-related illness risk in Grand Canyon National Park visitors to improve prevention

PONE-D-22-35008R1

Dear Dr. Buttke,

We’re pleased to inform you that your manuscript has been judged scientifically suitable for publication and will be formally accepted for publication once it meets all outstanding technical requirements.

Kind regards,

Mohammed Magdy Hamed

Academic Editor

PLOS ONE

Additional Editor Comments (optional):

Reviewers' comments:

Reviewer's Responses to Questions

**Comments to the Author**

1. If the authors have adequately addressed your comments raised in a previous round of review and you feel that this manuscript is now acceptable for publication, you may indicate that here to bypass the “Comments to the Author” section, enter your conflict of interest statement in the “Confidential to Editor” section, and submit your "Accept" recommendation.

Reviewer #3: All comments have been addressed

Reviewer #4: All comments have been addressed

2. Is the manuscript technically sound, and do the data support the conclusions?

Reviewer #3: Yes

Reviewer #4: Yes

3. Has the statistical analysis been performed appropriately and rigorously? 

Reviewer #3: Yes

Reviewer #4: I Don't Know

4. Have the authors made all data underlying the findings in their manuscript fully available?

Reviewer #3: Yes

Reviewer #4: Yes

5. Is the manuscript presented in an intelligible fashion and written in standard English?

Reviewer #3: Yes

Reviewer #4: Yes

6. Review Comments to the Author

Reviewer #3: (No Response)

Reviewer #4: Thank you for your efforts. The authors have addressed most comments.

I have three minor comments.

1- I feel the title could still be improved. "to improve prevention" could be removed or replaced.

2- Remove the references from the abstract.

3- Consider adding the main limitations of the study and accordingly the future work for research in the conclusions

7. PLOS authors have the option to publish the peer review history of their article (what does this mean?). If published, this will include your full peer review and any attached files.

Reviewer #3: No

Reviewer #4: No

---

## [Editor Report · Acceptance letter]

12 Jul 2023

PONE-D-22-35008R1 

Predicting climate-change induced heat-related illness risk in Grand Canyon National Park visitors 

Dear Dr. Buttke:

I'm pleased to inform you that your manuscript has been deemed suitable for publication in PLOS ONE. Congratulations! Your manuscript is now with our production department. 

Kind regards, 

on behalf of

Mr. Mohammed Magdy Hamed 

Academic Editor

PLOS ONE